# Estimation of Relative Chlorophyll Content in Spring Wheat Based on Multi-Temporal UAV Remote Sensing

Qiang Wu [1], Yongping Zhang [1,*], Zhiwei Zhao [1,*], Min Xie [1] and Dingyi Hou [2]

1   College of Agronomy, Inner Mongolia Agricultural University, Huhhot 010019, China
2   Chifeng Forest and Grassland Protection and Development Center, Chifeng 024005, China
*   Correspondence: imauzyp@163.com (Y.Z.); imauzzw@163.com (Z.Z.)

**Abstract:** Relative chlorophyll content (SPAD) is an important index for characterizing the nitrogen nutrient status of plants. Continuous, rapid, nondestructive, and accurate estimation of SPAD values in wheat after heading stage can positively impact subsequent nitrogen fertilization management strategies, which regulate grain filling and yield quality formation. In this study, the estimation of SPAD of leaf relative chlorophyll content in spring wheat was conducted at the experimental base in Wuyuan County, Inner Mongolia in 2021. Multispectral images of different nitrogen application levels at 7, 14, 21, and 28 days after the wheat heading stage were acquired by DJI P4M UAV. A total of 26 multispectral vegetation indices were constructed, and the measured SPAD values of wheat on the ground were obtained simultaneously using a handheld chlorophyll meter. Four machine learning algorithms, including deep neural networks (DNN), partial least squares (PLS), random forest (RF), and Adaptive Boosting (Ada) were used to construct SPAD value estimation models at different time from heading growth stages. The model's progress was evaluated by the coefficient of determination ($R^2$), root mean square error (RMSE), and mean absolute error (MAPE). The results showed that the optimal SPAD value estimation models for different periods of independent reproductive growth stages of wheat were different, with PLS as the optimal estimation model at 7 and 14 days after heading, RF as the optimal estimation model at 21 days after heading, and Ada as the optimal estimation model at 28 d after heading. The highest accuracy was achieved using the PLS model for estimating SPAD values at 14 d after heading (training set $R^2 = 0.767$, RMSE = 3.205, MAPE = 0.060, and $R^2 = 0.878$, RMSE = 2.405, MAPE = 0.045 for the test set). The combined analysis concluded that selecting multiple vegetation indices as input variables of the model at 14 d after heading stage and using the PLS model can significantly improve the accuracy of SPAD value estimation, provides a new technical support for rapid and accurate monitoring of SPAD values in spring wheat.

**Keywords:** wheat; machine learning; SPAD; vegetation indices





## 1. Introduction

Spring wheat is a major crop in northern China, and its growth and yield are critical for ensuring food security in the region. Chlorophyll, a pigment essential for photosynthesis in plants, has a strong influence on the nitrogen nutrition status, photosynthetic capacity, and yield of crops, and is a key parameter reflecting crop growth. Accurate and rapid estimation of chlorophyll levels can effectively assess the growth environment, water, and fertilizer management of crops, informing subsequent field management decisions and yield prediction [1–3]. Chemical methods are the traditional approach for measuring chlorophyll content, but they are laborious, destructive, and slow [4]. In addition, chlorophyll extracted from plant leaves is susceptible to decomposition by light, resulting in inaccurate measurement. The manual handheld chlorophyll meter, while faster than chemical methods, can only provide information on the chlorophyll content of a single leaf and does not account for the vertical heterogeneity within the canopy. Therefore, accurate measurement over a large area in time and space is not achievable with manual handheld chlorophyll meters.

Unmanned aerial vehicles (UAVs) have emerged as a promising remote sensing platform for obtaining physiological and biochemical traits of crops due to their mobility, flexibility, wide coverage, and high spatial and temporal resolution [5,6]. Spectral images acquired by UAVs in combination with algorithmic models have been shown to effectively monitor chlorophyll content [7–9]. While hyperspectral cameras have high inversion capabilities, they also have disadvantages, such as high cost, poor convenience, and complex operation processes. In contrast, multispectral images are less expensive and easier to control in flight and have equivalent inversion capabilities to hyperspectral images. Therefore, the use of UAV multispectral images for the acquisition and accurate inversion of crop growth parameters has great theoretical and practical value. Several studies have been conducted using UAV multispectral remote sensing technology to monitor the physical and chemical parameters of crops. Zhou et al. [10] developed a SPAD value inversion model for winter wheat using stepwise regression, principal component regression, and ridge regression. Niu et al. [11] employed two visible vegetation indices and four multispectral vegetation indices, along with stepwise regression and random forest regression methods, to estimate the SPAD values of winter wheat. Mao et al. [12] utilized two multispectral sensors with varying spectral response functions (Multiple Camera Array MCA and Sequoia) to obtain multispectral images of maize flowering under different nitrogen application levels and developed a more accurate estimation model by calculating vegetation indices and regressing them on the ground SPAD values.

The growth of wheat can be divided into three stages: the foundation stage, which begins at seed emergence and ends at stem elongation; the construction stage, which starts at the first node detachable from flowering and is a critical stage for yield; and the production stage, which begins at flowering and ends at ripening. Within the construction stage, there is also a period of nutritional and reproductive growth in parallel. After wheat heading, nutritional growth largely ceases, and the plant enters the independent reproductive growth stage. In previous research, the use of UAV multispectral imaging to estimate chlorophyll content has mostly focused on the flowering or early filling stages [13–15], with fewer studies examining the multi-temporal variation of wheat SPAD within the independent reproductive growth stage after heading. However, as the plant progresses through different growth stages, the optimal estimation model may change due to changes in the modeling data set. Therefore, this study employed separate modeling for different growth periods of wheat in order to achieve higher estimation accuracy.

The Hetao irrigation area, located in the northwestern part of China, is a region characterized by aridity and semi-aridity, and has limited water resources. In recent years, various water-saving irrigation patterns have been introduced in the region to replace conventional irrigation methods. These changes in irrigation patterns may impact the growth and development of wheat, which can be reflected in changes in canopy reflectance and SPAD values. Previous research on SPAD estimation of spring wheat in the Hetao irrigation area has not considered the effects of different irrigation modes on SPAD values. This study aims to develop an estimation model for SPAD values in spring wheat under both conventional and water-saving irrigation modes in the Hetao irrigation area, in order to improve the general applicability of the model for large-scale satellite remote sensing applications in the region.

## 2. Materials and Methods

### 2.1. Study Site and Experimental Design

In this study, two irrigation modes (conventional irrigation and water-saving irrigation) and six nitrogen fertilizer application rates were studied in a field trial. Multispectral images were collected using a UAV equipped with multispectral sensors at four time points (7, 14, 21, and 28 days after wheat heading) and were combined with ground measurements. Four machine learning regression models (DNN, PLS, RF, and Ada) were used to determine the optimal period and model for estimating SPAD values after the heading stage of spring

wheat in the Hetao irrigation area of Inner Mongolia. The results of this study provide theoretical support for remote sensing monitoring of SPAD values in this region.

This study was conducted at Wuyuan Agricultural Technology Extension Center (107°35′ N, 40°30′50″ E, elevation 1028 m a.s.l.), located in Bayannur City, Inner Mongolia, China, during 2021 (location is shown in Figure 1). The region has a temperate continental monsoon climate. The soil type at the experimental site was loam, with baseline fertility level of organic matter 17.65 g/kg, alkaline nitrogen 57.45 mg/kg, available phosphorus 26.83 mg/kg, available potassium 152.42 mg/kg, and pH = 7.32. The spring wheat cultivar "Yongliang 4" was selected for the study. The experiment used a split-plot design, with irrigation as the main plot and nitrogen (N) application as the subplot. There were two irrigation modes: conventional irrigation (four times at tillering stage, jointing stage, flowering stage, and early grain filling stage) and water-saving irrigation (two times at jointing stage and flowering stage), each with a volume of 900 m³/ha using flood irrigation. The N application subplot had six levels: CK (no fertilizer), N0 (0 kg/ha), N1 (75 kg/ha), N2 (150 kg/ha), N3 (225 kg/ha), N4 (300 kg/ha). The experiment had a total of 12 treatments with three replications, resulting in 36 experimental plots of 42 m² each. Phosphorus fertilizer was applied as a base fertilizer at sowing, and the sowing rate was set at 375 kg/ha. Rainfall and temperature data are shown in Figure 2.

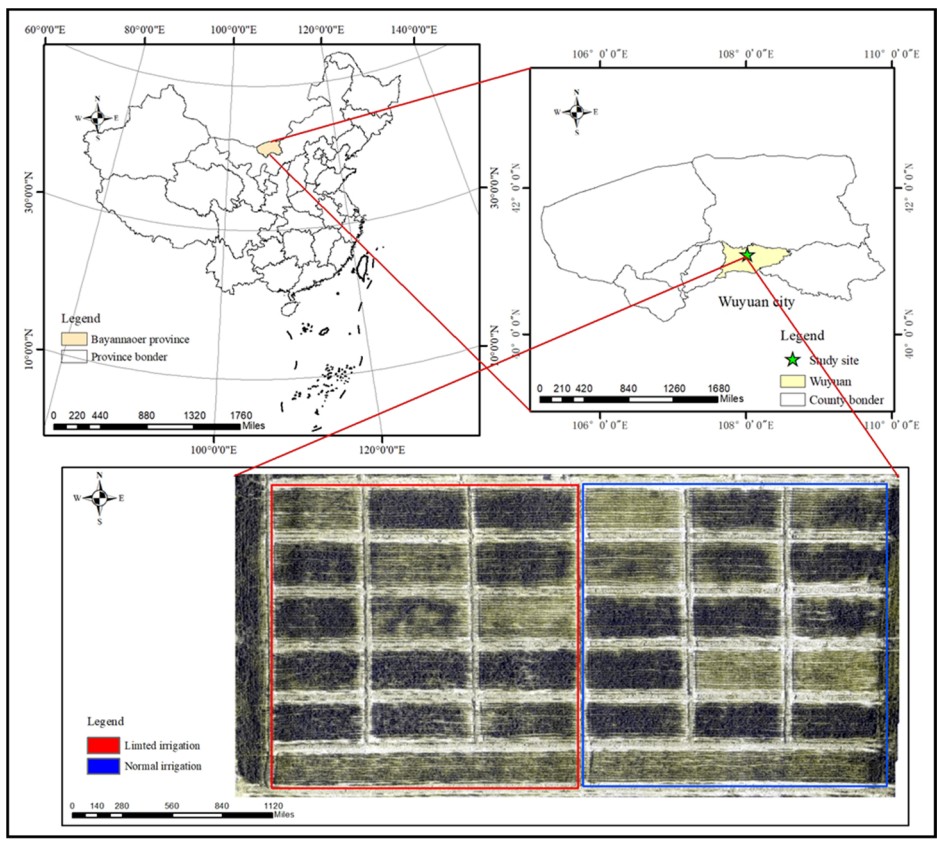

**Figure 1.** Geographical location of the experimental site. Red frame is 2W treatment; Blue frame is 4W treatment.

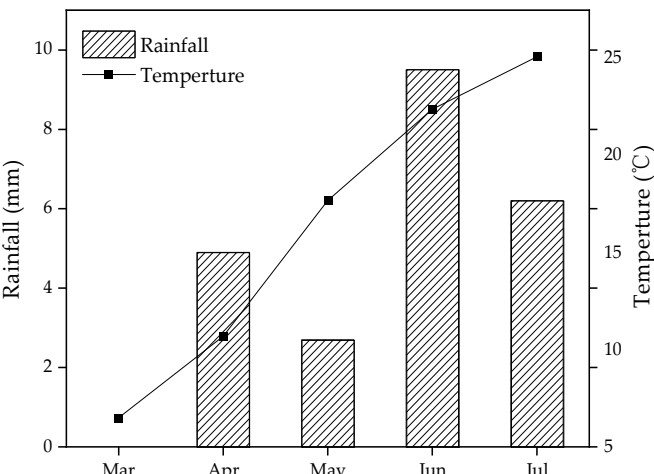

**Figure 2.** Rainfall and temperature of experimental site. Mar: March, Apr: April, Jun: June, Jul: July.

### 2.2. UAV Multispectral Data Acquisition and Processing

Multispectral image data were collected at 7, 14, 21, and 28 days after heading stage of the wheat plant using a DJI Phantom 4 multispectral drone (Da-Jiang Innovations, Shenzhen, China).The drone (P4M, Figure 3) integrates five multispectral sensors (blue B: $450 \pm 16$ nm, green G: $560 \pm 16$ nm, red R: $650 \pm 16$ nm, red edge RE: $730 \pm 16$ nm 16 nm, near infrared NIR: $840 \pm 26$ nm) and one RGB visible light sensor. To avoid hotspot phenomenon in the images, the images were acquired between 9:00 and 11:00 a.m. on clear and windless days, with the takeoff location fixed and kept consistent each time. Before takeoff, the UAV was manually placed directly above the three reflectivity gray plates of 20%, 40%, and 60%, and reflectivity plate photos were taken. The flight path was automatically planned by DJI GS Pro after calculating the current solar azimuth, with a flight altitude of 30 m, a heading overlap of 85%, and a collateral overlap of 80%. The D-RTK 2 high-precision GNSS mobile station was used to assist the positioning of the UAV and improve the positioning accuracy of the UAV itself. After the flight, DJI Terra was used to perform radiometric correction of the images acquired during the mission, followed by image stitching to obtain a single-band reflectivity orthophoto.

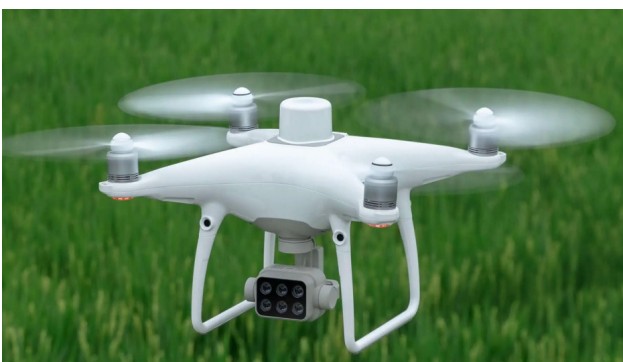

**Figure 3.** A photo for Phantom 4 multispectral.

### 2.3. Construction and Selection of Spectral Indices

The reflectance of each treatment plot was extracted by the zonal statistics function of ENVI, and the following vegetation indices (VIs, Vegetable Indices) were calculated (Table 1).

**Table 1.** Vegetation indices and calculation method.

| Index Name | Calculation Formula | References |
|---|---|---|
| Leaf chlorophyll index | $LCI = (R_{nir} - R_{rededge})/(R_{nir} + R_{red})$ | [16] |
| Difference vegetation index | $DVI = R_{nir} - R_{red}$ | [16] |
| Enhanced Vegetation Index | $EVI = 2.5 \times (R_{nir} - R_{red})/(R_{nir} + 6 \times R_{red} - 7.5 \times R_{blue} + 1)$ | [16] |
| Green Normalized Difference Vegetation | $GNDVI = (R_{nir} - R_{green})/(R_{nir} + R_{green})$ | [17] |
| Ratio Between NIR and Green Bands | $VI_{(nir/green)} = R_{nir}/R_{green}$ | [18] |
| Ratio Between NIR and Red Bands | $VI_{(nir/red)} = R_{nir}/R_{red}$ | [19] |
| Ratio Between NIR and Red Edge Bands | $VI_{(nir/rededge)} = R_{nir}/R_{rededge}$ | [20] |
| Napierian Logarithm of The Red Edge | $\ln_{RE} = 100 \times (\ln_{nir} - \ln_{red})$ | [21] |
| Modified Soil-Adjusted Vegetation Index 1 | $MSAVI1 = (1 + L)\left(\frac{R_{nir} - R_{red}}{R_{nir} + R_{red} + L}\right)(L = 0.1)$ | [22] |
| Modified Soil-Adjusted Vegetation Index 2 | $MSAVI2 = R_{nir} + 0.5 - \sqrt{(2 \times R_{nir} + 1)^2 - 8 \times (R_{nir} - R_{red})}/2$ | [22] |
| Optimized Soil-Adjusted Vegetation Index | $OSAVI = (1 + 0.16) \times \frac{(R_{nir} - R_{red})}{(R_{nir} + R_{red} + 0.16)}$ | [23] |
| Modified Triangular Vegetation Index 2 | $MTVI2 = \frac{1.5 \times [1.2 \times (R_{nir} - R_{green}) - 2.5 \times (R_{red} - R_{green})]}{\sqrt{(2 \times R_{nir} + 1)^2 - (6 \times R_{nir} - 5 \times \sqrt{R_{red}}) - 0.5}}$ | [24] |
| Normalized Difference Red Edge Index | $NDRE = \frac{(R_{nir} - R_{rededge})}{(R_{nir} + R_{rededge})}$ | [25] |
| Normalized Difference Vegetation Index | $NDVI = \frac{(R_{nir} - R_{red})}{(R_{nir} + R_{red})}$ | [26] |
| Modified Simple Radio | $MSR = (R_{nir} - R_{red} - 1)/(\sqrt{R_{nir} + R_{red}} + 1)$ | [27] |
| Soil-Adjusted Vegetation Index | $SAVI = \frac{(R_{nir} - R_{red})}{(R_{nir} + R_{red} + 0.5)} \times (1 + 0.5)$ | [28] |
| Simplified Canopy Chlorophyll Content Index | $SCCCI = \frac{NDRE}{NDVI}$ | [29] |
| Modified Chlorophyll Absorption Reflectance Index | $MCARI = (R_{rededge} - R_{red} - 0.2 \times (R_{rededge} - R_{green})) \times \left(\frac{R_{rededge}}{R_{red}}\right)$ | [30] |
| Modified Chlorophyll Absorption Reflectance Index 2 | $MCARI2 = 1.5 \times \frac{(2.5 \times (R_{nir} - R_{rededge}) - 1.3 \times (R_{nir} - R_g))}{(2 \times (R_{nir} + 1)^2 - (6 \times R_{nir} - 5 \times (R_{red})^2) - 0.5)}$ | [31] |
| Transformed Chlorophyll Absorption Reflectance Index | $TCARI = 3 \times \left((R_{rededge} - R_{red}) - 0.2 \times (R_{rededge} - R_{green}) \times \left(\frac{R_{rededge}}{R_{red}}\right)\right)$ | [32] |
| Normalized Difference Index | $NDI = \frac{(R_{nir} - R_{rededge})}{(R_{nir} + R_{red})}$ | [33] |
| Red-Edge Chlorophyll Index 1 | $Cl1 = \frac{R_{nir}}{R_{rededge}} - 1$ | [34] |
| Red-Edge Chlorophyll Index 2 | $Cl2 = \frac{R_{rededge}}{R_{green}} - 1$ | [35] |
| Structure-Insensitive Pigment Index | $SIPI = \frac{(R_{nir} - R_{blue})}{(R_{nir} + R_{red})}$ | [36] |
| TCARI/OSAVI | $\frac{TCARI}{OSAVI}$ | [31] |
| MCARI/OSAVI | $\frac{MCARI}{OSAVI}$ | [31] |

### 2.4. Ground Data Acquisition and Processing

During images collection by the UAV, five wheat plants were selected in each plot according to the "five-point sampling method", and the SPAD values of leaf tip, leaf middle, and leaf base of the flag leaf of the wheat plant were measured using a SPAD 502Plus chlorophyll meter (Konica Minolta, Tokyo, Japan). The average value was taken as the SPAD value of the plant.

### 2.5. Construction of Regression Model

In this study, four regression models were implemented in Python for the estimation of SPAD values as follows.

Deep neural networks (DNN) is a neural network containing multiple hidden layers (at least 3 layers), which has more hidden layers and a stronger fitting ability compared to traditional neural networks.

Partial least squares (PLS) draws on the advantages of statistical methods, such as correlation analysis, principal component analysis, and multiple linear regression, and is widely used in hyperspectral inversion estimation with high correlation between independent variables because of its strong ability to remove autocorrelation between features.

Random forest regression (RF) is a machine learning algorithm that uses multiple decision trees to train and predict samples and has strong anti-interference ability. It also has the advantages of fast training speed and no processing of input data.

Adaptive Boosting (Ada) is one of the representative algorithms of Boosting in integrated learning, which mainly changes to obtain different test samples by controlling the weights of sample distribution.

*2.6. Segmentation of Dataset and Accuracy Evaluation*

The samples of each period were randomly divided into training set and test set at the ratio of 7:3, and K-fold cross validation (K = 5) was used to optimize the model. Five-fold cross-validation involves dividing the original training set into 5 groups, using each subset of data as a validation set in turn, and using the remaining 4 subsets of data as the training set. The results from K groups are then summed and averaged to reduce the error of the training set and improve the generalization ability of the model by avoiding the inclusion of test data during the training process.

The accuracy of the model is evaluated by three metrics: the coefficient of determination ($R^2$), the root means square error (RMSE), and the mean absolute prediction error (MAPE). $R^2$ is used to indicate the degree of fit between the estimated and measured values, with a value closer to 1 indicating a higher accuracy of the model fit. RMSE reflects the deviation of the estimated value from the measured value, with a smaller value indicating a higher accuracy of the model fit. MAPE is the average of absolute errors, which more accurately reflects the actual errors in the prediction value.

$$R^2 = 1 - \frac{\sum_{i=1}^{n}(y_i - \hat{y}_i)^2}{\sum_{i=1}^{n}(y_i - \bar{y}_i)^2} \tag{1}$$

$$RMSE = \sqrt{\frac{\sum_{i=1}^{n}(y_i - \hat{y}_i)^2}{n}} \tag{2}$$

$$MAPE = \frac{100\%}{n} \sum_{i=1}^{n} \left| \frac{\hat{y}_i - y_i}{y_i} \right| \tag{3}$$

$y_i$ is the observed value, $\bar{y}_i$ is the mean of the observed values, $\hat{y}_i$ is the model predicted value, and *n* is the number of samples.

## 3. Results

*3.1. Basic Statistical Information of Measured SPAD Values*

The SPAD values of different treatments are shown in Table 2. As the reproductive process advances, different N treatments under the 2W treatment showed a trend of increasing then decreasing, the maximum SPAD value appeared 21 days after heading, and the maximum SPAD value of different N treatments under the 4W treatment appeared 7 days after heading. With the increase in nitrogen application, the SPAD values after different heading periods showed a trend of increasing and then decreasing.

**Table 2.** Basic statistics of the field measurements at different stages.

| Irrigation | N Treatment | 7 d | 14 d | 21 d | 28 d |
|---|---|---|---|---|---|
| 2W | N0 | 41.99 bcd | 43.42 bcd | 43.11 d | 36.33 bc |
| | N5 | 46.90 abc | 47.75 abc | 47.05 bcd | 42.99 ab |
| | N10 | 48.42 abc | 50.23 ab | 51.89 ab | 43.82 ab |
| | N15 | 50.10 ab | 51.05 a | 55.50 a | 45.29 a |
| | N20 | 48.58 abc | 46.05 abcd | 51.53 abc | 46.51 a |
| | CK | 39.95 cd | 36.26 de | 33.46 e | 14.73 e |
| 4W | N0 | 41.70 bcd | 35.75 ef | 28.47 e | 10.47 e |
| | N5 | 46.61 abc | 42.75 bcde | 45.09 cd | 24.35 d |
| | N10 | 47.97 abc | 41.80 cde | 45.81 bcd | 29.48 cd |
| | N15 | 51.15 a | 46.77 abcd | 50.62 abc | 34.55 c |
| | N20 | 41.65 bcd | 45.75 abcd | 46.13 bcd | 26.15 d |
| | CK | 36.29 cd | 29.63 f | 31.37 e | 12.87 e |

Alphabets within columns followed by the same letter are statistically insignificant at the 0.05 level.

### 3.2. Correlation Analysis of SPAD Values and Vegetation Indices

The correlation coefficients between SPAD values and each vegetation index at different periods after heading are shown in Table 3. The highest correlation coefficient was observed at 7 days after heading, followed by 21 days. Except for 7 days after heading, the correlation coefficient for the rest of the period showed 2W < 4W. Under the 2W treatment, the vegetation indices with the highest correlation coefficients at 7, 14, 21, and 28 days after tapping were MCARI2, MSAVI2, SCCCI, and MCARI. The highest correlation coefficients under 4W were MCARI2, MSAVI1, CL1, and DVI.

**Table 3.** Correlation coefficients between spectral vegetation indices and SPAD.

| Indices | 7 d | | 14 d | | 21 d | | 28 d | |
|---|---|---|---|---|---|---|---|---|
| | 2W | 4W | 2W | 4W | 2W | 4W | 2W | 4W |
| DVI | 0.912 | 0.867 | 0.650 | 0.867 | 0.701 | 0.830 | 0.708 | 0.802 |
| EVI | 0.908 | 0.866 | 0.695 | 0.871 | 0.708 | 0.856 | 0.727 | 0.798 |
| NDVI | 0.842 | 0.815 | 0.790 | 0.871 | 0.703 | 0.867 | 0.740 | 0.790 |
| GNDVI | 0.882 | 0.825 | 0.776 | 0.870 | 0.741 | 0.903 | 0.741 | 0.779 |
| NDRE | 0.912 | 0.846 | 0.739 | 0.863 | 0.764 | 0.925 | 0.749 | 0.774 |
| LCI | 0.912 | 0.846 | 0.739 | 0.863 | 0.764 | 0.925 | 0.749 | 0.774 |
| OSAVI | 0.879 | 0.847 | 0.745 | 0.873 | 0.714 | 0.876 | 0.734 | 0.799 |
| VI(NIR/G) | 0.893 | 0.867 | 0.671 | 0.830 | 0.765 | 0.919 | 0.742 | 0.756 |
| VI(NIR/R) | 0.842 | 0.863 | 0.637 | 0.819 | 0.761 | 0.918 | 0.726 | 0.754 |
| VI(NIR/RE) | 0.917 | 0.863 | 0.698 | 0.849 | 0.765 | 0.926 | 0.748 | 0.766 |
| lnRE | 0.856 | 0.844 | 0.735 | 0.853 | 0.747 | 0.916 | 0.742 | 0.782 |
| MSAVI1 | 0.870 | 0.840 | 0.759 | 0.873 | 0.713 | 0.875 | 0.736 | 0.797 |
| MSAVI2 | 0.729 | 0.706 | 0.815 | 0.855 | 0.589 | 0.796 | 0.734 | 0.719 |
| MTVI2 | 0.895 | 0.868 | 0.677 | 0.863 | 0.711 | 0.859 | 0.722 | 0.801 |
| MSR | 0.915 | 0.859 | 0.640 | 0.866 | 0.692 | 0.801 | 0.679 | 0.793 |
| SAVI | 0.898 | 0.860 | 0.706 | 0.872 | 0.712 | 0.867 | 0.727 | 0.802 |
| SCCCI | 0.924 | 0.840 | 0.723 | 0.864 | 0.772 | 0.925 | 0.231 | −0.652 |
| MCARI | −0.824 | −0.782 | −0.785 | −0.854 | −0.739 | −0.881 | −0.783 | 0.463 |
| MCARI2 | 0.929 | 0.870 | 0.663 | 0.853 | 0.764 | 0.910 | 0.746 | 0.750 |
| TCARI | −0.817 | −0.850 | −0.638 | −0.830 | −0.766 | −0.921 | −0.562 | 0.682 |
| NDI | 0.902 | 0.840 | 0.755 | 0.866 | 0.758 | 0.920 | 0.749 | 0.779 |
| CL1 | 0.917 | 0.863 | 0.698 | 0.849 | 0.765 | 0.926 | 0.748 | 0.766 |
| CL2 | 0.869 | 0.850 | 0.710 | 0.840 | 0.759 | 0.910 | 0.743 | 0.763 |
| SIPI | 0.834 | 0.809 | 0.779 | 0.865 | 0.713 | 0.867 | 0.735 | 0.788 |
| TCARI/OSAVI | −0.826 | −0.841 | −0.689 | −0.846 | −0.753 | −0.919 | −0.766 | −0.618 |
| MCARI/OSAVI | −0.826 | −0.786 | −0.797 | −0.870 | −0.695 | −0.847 | −0.757 | −0.742 |

### 3.3. Model Development and Evaluation

3.3.1. Estimation of SPAD Values after Heading 7 d

Table 4 and Figure 4 shows the evaluation of the accuracy of the SPAD estimation models for 7 d after wheat heading stage. the $R^2$ of the training set and the test set of the four models are above 0.70, with the accuracy of the training set higher accuracy than the test set. Using the accuracy of the test set as the evaluation criterion of the models, the accuracy of the models was found to be in the following order: PLS > RF > Ada > DNN. The $R^2$, RMSE, and MAPE values of the test set of the PLS model were 0.762, 3.048, and 0.052, respectively, which were 7.32%, 2.28%, and 7.17% higher than the $R^2$ of the RF, Ada, and DNN models, respectively. RMSE decreased by 9.25%, 3.33%, and 9.20%, and MAPE decreased by 21.25%, 0%, and 7.69%. Based on these results, it can be concluded that the PLS model had the highest accuracy and stability in estimating the SPAD values of wheat 7 d after heading stage.

**Table 4.** Accuracy assessment of different estimation models at 7 d after heading.

| Model | Training Set | | | Test Set | | |
|---|---|---|---|---|---|---|
| | $R^2$ | RMSE | MAPE | $R^2$ | RMSE | MAPE |
| DNN | 0.754 | 2.784 | 0.048 | 0.710 | 3.359 | 0.063 |
| PLS | 0.786 | 2.595 | 0.043 | 0.762 | 3.048 | 0.052 |
| RF | 0.957 | 1.169 | 0.021 | 0.745 | 3.153 | 0.052 |
| Ada | 0.968 | 0.829 | 0.014 | 0.711 | 3.357 | 0.056 |

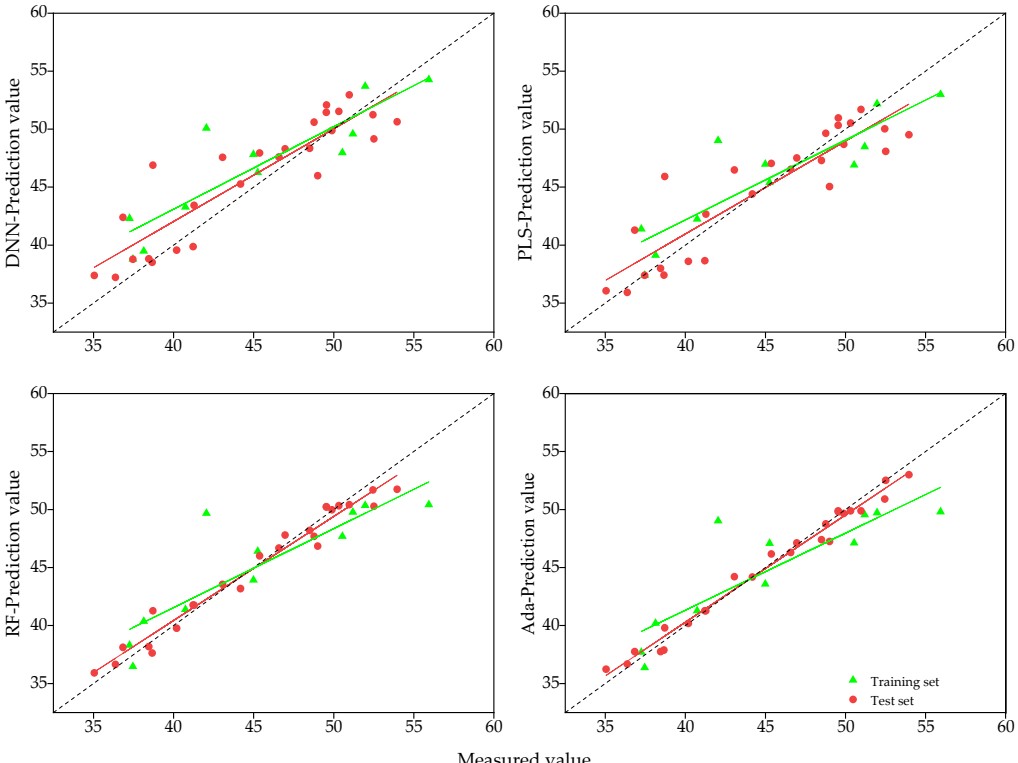

**Figure 4.** Calibration and validation results of different estimation models at 7 d after heading.

### 3.3.2. Estimation of SPAD Values after Heading 14 d

The accuracy of the SPAD estimation models for 14 d after wheat heading stage was evaluated and presented in Table 5 and Figure 5. The $R^2$ values for the training and test sets of the four models are above 0.75, and the accuracy of the training set of the DNN and PLS models was lower than that of the test set, while the accuracy of the training set of the RF and Ada models was higher than that of the test set. When using the accuracy of the test set as the evaluation criterion for the models, the accuracies of different models were found to be PLS > Ada > RF > DNN in descending order. The $R^2$, RMSE, and MAPE values for the test set of PLS model were 0.878, 2.405, and 0.045, which were 12.13%, 10.72%, and 7.33% higher than the $R^2$ values of the RF, Ada, and DNN models, respectively, and the RMSE is reduced by 25.05%, 23.41%, and 18.28%, and the MAPE decreased by 44.44%, 53.33%, and 35.56%. The comprehensive analysis concluded that the PLS model had the highest accuracy and stability in estimating the SPAD values of wheat 14 d after heading stage.

**Table 5.** Accuracy assessment of different estimation models at 14 d after heading.

| Model | Training Set | | | Test Set | | |
|---|---|---|---|---|---|---|
| | $R^2$ | RMSE | MAPE | $R^2$ | RMSE | MAPE |
| DNN | 0.716 | 3.538 | 0.067 | 0.783 | 3.209 | 0.065 |
| PLS | 0.767 | 3.205 | 0.060 | 0.878 | 2.405 | 0.045 |
| RF | 0.924 | 1.835 | 0.036 | 0.793 | 3.140 | 0.069 |
| Ada | 0.934 | 1.711 | 0.029 | 0.818 | 2.943 | 0.061 |

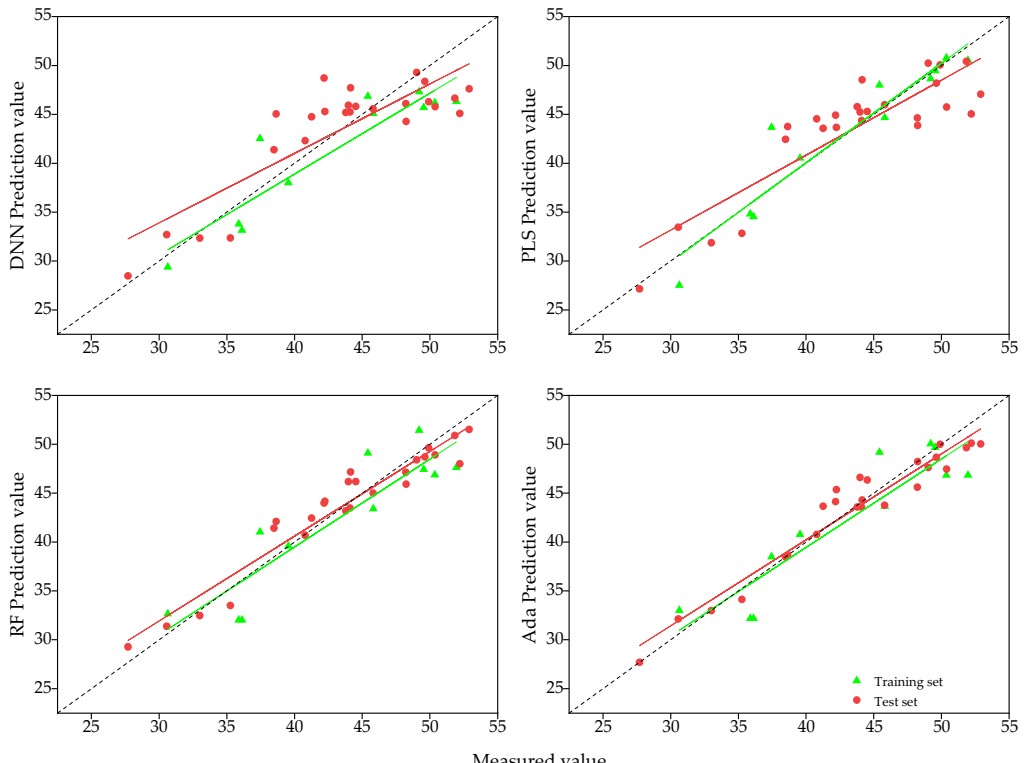

**Figure 5.** Calibration and validation results of different estimation models at 14 d after heading.

### 3.3.3. Estimation of SPAD Values after Heading 21 d

The accuracy of SPAD estimation models for 21 d after wheat heading stage was shown in Table 6 and Figure 6. The $R^2$ of training set and test set of all four models were above 0.65, with the training set accuracy of the DNN and PLS models higher than the test set, while the training set accuracy of the RF and Ada models were lower than the test set. The accuracy of different models, from largest to smallest, was DNN > RF > Ada > PLS. The $R^2$ value of the DNN test set model was 0.737, the RMSE was 4.806, and the MAPE was 0.086, which were 12.18%, 1.80%, and 5.74% higher than the $R^2$ values of the PLS, RF, and Ada models, respectively. The RMSE was 12.41%, 2.42%, −11.02%, and the MAPE decreased by 28.33%, 18.10%, and −4.88%. Based on this analysis, it can be concluded that the DNN model was the most accurate and stable in estimating the SPAD values of wheat 21 d after heading stage.

**Table 6.** Accuracy assessment of different estimation models at 21 d after heading.

| Model | Training Set | | | Test Set | | |
|---|---|---|---|---|---|---|
| | $R^2$ | RMSE | MAPE | $R^2$ | RMSE | MAPE |
| DNN | 0.881 | 2.922 | 0.044 | 0.737 | 4.806 | 0.086 |
| PLS | 0.777 | 4.009 | 0.080 | 0.657 | 5.487 | 0.120 |
| RF | 0.678 | 4.815 | 0.097 | 0.724 | 4.925 | 0.105 |
| Ada | 0.684 | 5.028 | 0.092 | 0.697 | 4.329 | 0.082 |

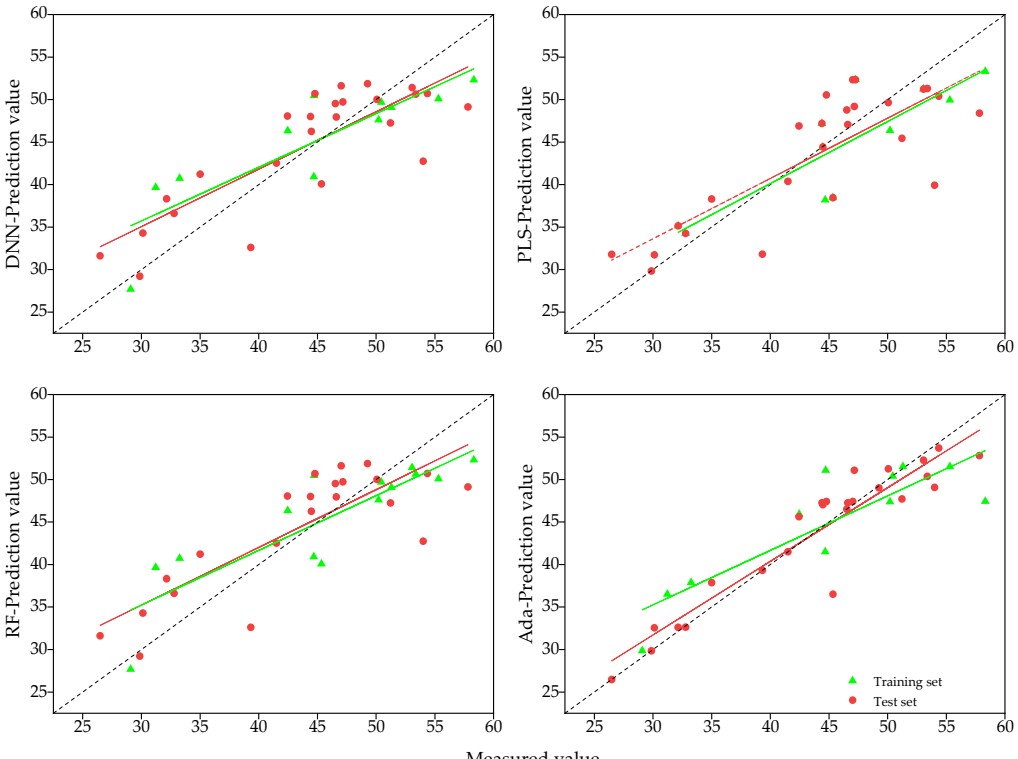

**Figure 6.** Calibration and validation results of different estimation models at 21 d after heading.

### 3.3.4. Estimation of SPAD Values after Heading 28 d

The accuracy of the SPAD estimation model for 28 d after the wheat heading stage was evaluated and presented in Table 7 and Figure 7. The $R^2$ values for the training and test set of all four models are above 0.65, and the accuracy of the training set is higher than the test set for all models except the DNN model. When using the test set accuracy as the evaluation criterion, the Ada model was found to be the most accurate, followed by RF, PLS, and DNN in descending order. The $R^2$ value for the Ada model was 0.815, the RMSE was 5.904, and the MAPE was 0.237, which were 20.38%, 14.63%, and 14.47% higher, respectively, than those of the DNN, PLS, and RF models. The RMSE was reduced by 62.77%, 60.67%, and 60.59%, and the MAPE was reduced by 22.80%, 20.74%, and 31.30% for the Ada model compared to the other models. Overall, the Ada model was found to have the highest accuracy and stability in estimating the SPAD values of wheat 28 d after heading stage.

**Table 7.** Accuracy assessment of different estimation models at 28 d after heading.

| Model | Training Set | | | Test Set | | |
|---|---|---|---|---|---|---|
| | R$^2$ | RMSE | MAPE | R$^2$ | RMSE | MAPE |
| DNN | 0.691 | 7.054 | 0.262 | 0.677 | 7.801 | 0.307 |
| PLS | 0.713 | 6.803 | 0.299 | 0.711 | 7.383 | 0.299 |
| RF | 0.926 | 3.442 | 0.115 | 0.712 | 7.368 | 0.345 |
| Ada | 0.971 | 2.168 | 0.067 | 0.815 | 5.904 | 0.237 |

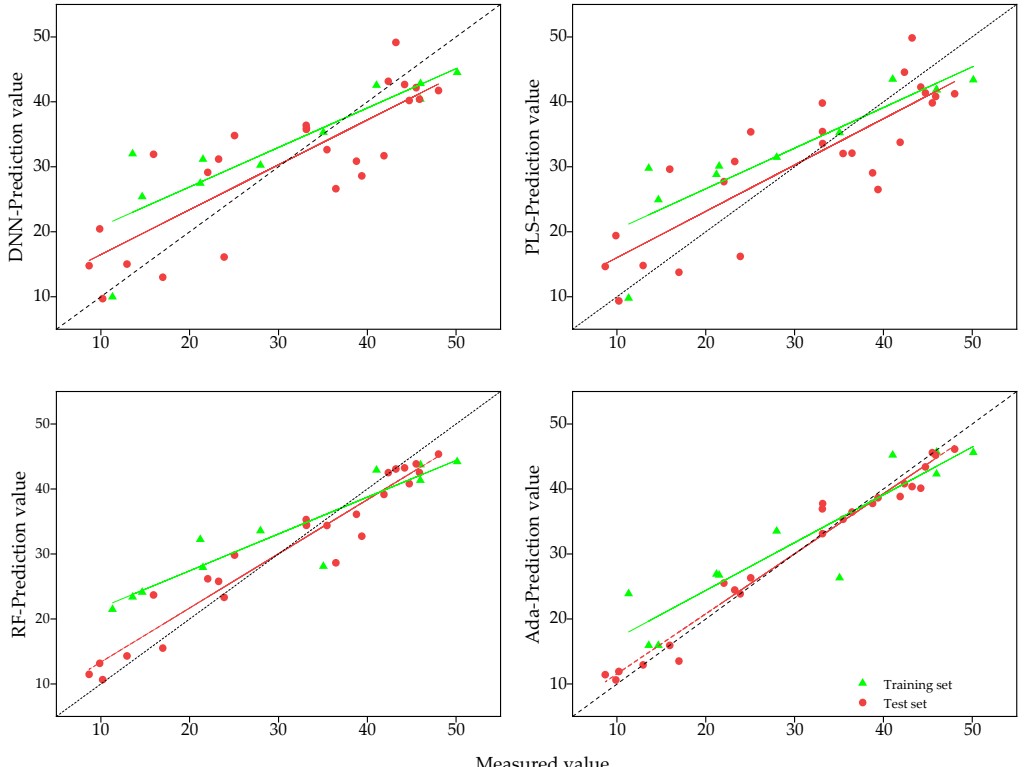

**Figure 7.** Calibration and validation results of different estimation models at 28 d after heading.

### 3.4. Comparison of Accuracy of Four Estimation Models at Different Growth Stages

As the growth process progressed, the accuracy of the four models demonstrated an overall trend of increasing then decreasing and then increasing again. The optimal model for estimating SPAD values varied depending on the growth stage (Figure 8). The model with the highest R$^2$ value at 7 and 14 days after heading was PLS, the model with the highest R$^2$ value at 21 d after heading was RF, and the model with the highest R$^2$ value at 28 d after heading was Ada. The model with the lowest RMSE value at 7 and 14 days after heading was PLS, and the model with the lowest RMSE value at 21 and 28 days after heading was Ada. The model with the lowest MAPE value at 7 days after heading was PLS and RF, the lowest at 14 days after heading was PLS, and the lowest at 21 and 28 days after heading was Ada.

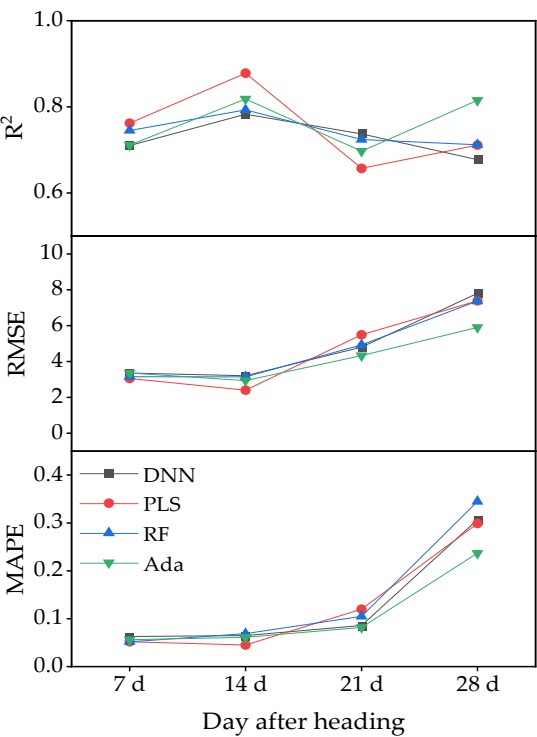

**Figure 8.** Comparison of estimation accuracy of four models test set at different growth stages.

## 4. Discussion

Leaf SPAD value is an important indicator for characterizing the nitrogen nutrition status of plants [37], and several studies have been conducted to estimate the SPAD value of crop plants using the unmanned aerial vehicle (UAV) remote sensing technology platform in order to diagnose crop nitrogen nutrition and provide a reference basis for subsequent field nitrogen fertilization management [38,39]. Fast access to the growth conditions of crops in farmland is an important aspect of smart agriculture for fast sensing and intelligent decision making. Although there are few studies that focus on the independent reproductive growth stage (from heading to maturity) of wheat, monitoring leaf SPAD values during this stage and adopting appropriate water and fertilizer management can coordinate the development of population and individual main stem and tillers, and have significant regulatory effects on late grain filling, final yield, and grain quality formation. In this study, multispectral images were acquired every 7 days after the wheat heading stage using an unmanned aerial vehicle, and vegetation indices were extracted to construct separate SPAD estimation models for wheat.

Vegetation indices have been found to be correlated with agronomic traits making them an important alternative to traditional agronomic parameters [40]. This study demonstrated that the measured SPAD values of wheat at various growth stages had strong correlations and linear sensitivity with most vegetation indices. However, this study also found that the correlation between the corresponding SPAD values and the vegetation indices gradually decreased as the reproductive process progressed. This may be due to the fact that during the nutritional growth stage of wheat, most of the absorbed and accumulated nitrogen is stored in nutritional organs, such as leaves [41], leading to a higher spectral reflectance sensitivity of the canopy. During the independent reproductive growth stage, nitrogen is gradually transferred from nutritional organs to the spike, the leaves senesce and degrade [42], resulting in a decrease in spectral reflectance sensitivity.

The spectral characteristics of the vegetation canopy are influenced by numerous physical and biochemical variables, which exhibit different behaviors at different growth stages [43,44]. As a result, the accuracy of the estimation models constructed based on the extraction of vegetation indices from spectral reflectance varies across fertility periods.

Previous research has shown that the accuracy of estimation models tends to decrease at the heading, flowering, and filling stages of winter wheat [45]. Another study found that the accuracy of three regression models for estimating SPAD values in winter wheat increased, then decreased, and then increased again during the four growth periods from nodulation to flowering [10]. In the present study, the accuracy of the model constructed at the four growth stages showed relatively consistent performance, except for the SPAD value estimation model at 21 d of heading stage. The width of a violin plot represents the probability density of the data. As shown in Figure 9c, the distribution of spectral indices at 21 d after heading displayed an obvious concentration and an oversaturation phenomenon, leading to a decreased sensitivity of vegetation indices and subsequently increased error and decreased model accuracy in the estimation process. In contrast, after 28 d of heading stage, plants treated with low levels of nitrogen exhibited senescence, chlorophyll decomposition, and decreased chlorophyll content, while those treated with high levels of nitrogen maintained high chlorophyll content, which to some extent weakened the oversaturation phenomenon and thus improved model accuracy.

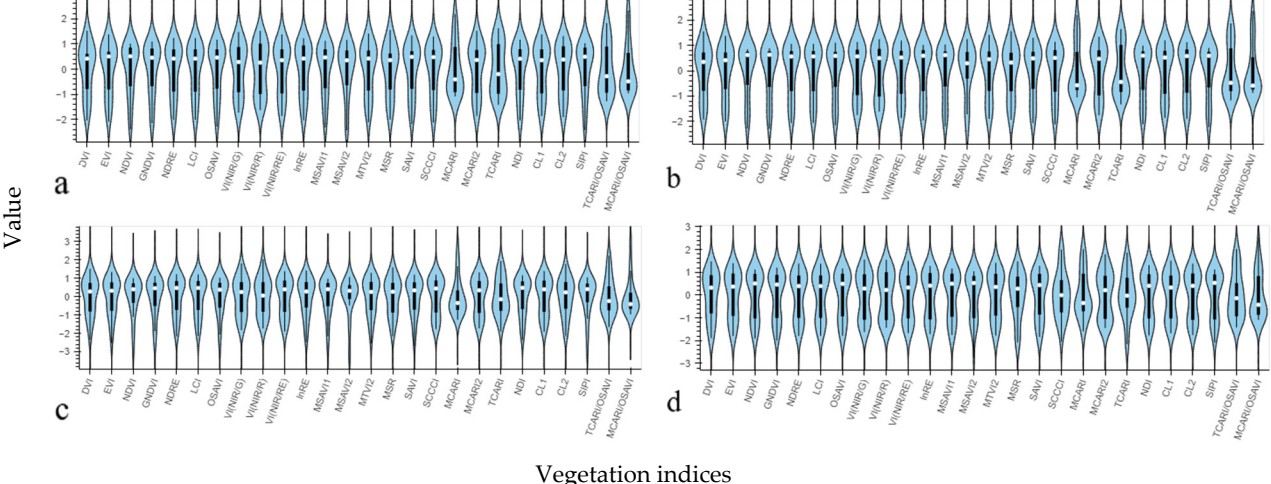

**Figure 9.** Violin plots of the spectral indices. (**a**–**d**) are violin plots of vegetation indices at 7, 14, 21, and 28 days after heading, respectively.

Compared with using a single vegetation index as the model input variable, the estimation method based on multiple vegetation indices can improve the model accuracy to a certain extent. In a multi-year study by Kooistra et al. [46] to estimate chlorophyll content of potato leaves, the regression model obtained with a single vegetation index as an input variable had a maximum $R^2$ of 0.641. Multiple vegetation indices were used in this study to jointly participate in the estimation, and the model with the lowest precision among the four periods, $R^2 = 0.657$, was also above this precision. Yuan et al. [47] found in their study on estimating chlorophyll content of plant seedlings that most vegetation indices, although significantly correlated with SPAD values, had low $R^2$ for regression models with a single vegetation index as the explanatory variable, and the accuracy of the regression models was significantly improved when multiple vegetation indices were used as explanatory variables. In this study, all four regression models based on 26 vegetation indices demonstrated good performance, with test set $R^2$ ranging from 0.657 to 0.878, RMSE ranging from 2.934 to 7.801, and MAPE ranging from 0.045 to 0.345. This suggests that the inclusion of multiple vegetation indices in the regression models allowed for the incorporation of more valid spectral information, resulting in improved estimation accuracy.

The accuracy of different models for estimating SPAD values of spring wheat varied significantly across different data sets. An appropriate estimation model can effectively capture the relationship between vegetation indices and SPAD values and maintain stable model structure and parameters. Previous studies have shown the effectiveness of machine

learning algorithms in plant nutrition diagnosis [48,49], and it is important to identify the best model for estimating SPAD at different stages of fertility in spring wheat after heading stage. In this study, the optimal model was PLS at 7 and 14 days after heading, RF at 21 d after heading, and Ada at 28 d after heading. The $R^2$ of the training set of the Ada and RF models at 7, 14, and 28 days was significantly higher than that of the test set, and showed a significant overfitting phenomenon, probably due to the high autocorrelation among the 26 vegetation indices of the input, which was influenced by the multicollinearity of the input feature variables. The regression modeling method of PLSR converts a set of highly correlated independent variables into a set of mutually independent, non-linearly related principal component variables by extracting principal components in the process of establishing regressions, which can effectively capture most of the information of the original data and eliminate the covariance among vegetation indices, resulting in the best estimation accuracy of all four models being obtained at 7 d and 14 d after sampling. In the case of deep learning models, a large amount of diverse data is typically required for model training in order to understand the relationship between data and estimates, and the number of samples in this study was not sufficient to support a deep learning network with multiple hidden layers, leading to a low level of estimation accuracy for the DNN model. However, the estimation accuracy ($R^2$) for all four periods was found to be greater than 0.677, indicating the strong potential for the DNN model to be used for estimation. This finding is in line with the results of Liu et al. [50].

## 5. Conclusions

The optimal SPAD estimation models were different for the four periods after the spring wheat heading stage, with PLS as the optimal model at 7 and 14 d after heading stage, RF at 21 d after heading stage, and Ada at 28 d after heading stage, where the highest accuracy was achieved by using the PLS model to estimate SPAD values at 14 d after heading stage (training set $R^2$ = 0.767, RMSE = 3.205, MAPE = 0.060, and test set $R^2$ = 0.878, RMSE = 2.405, MAPE = 0.045).

Further studies could include validation of the model's performance on additional datasets, assessment of its practical usefulness and potential adoption by farmers through field trials, integration with precision agriculture tools and technologies, such as drones or sensors, and the optimization of fertilization and irrigation practices to improve wheat yield and quality.

**Author Contributions:** Conceptualization, Q.W. and Y.Z.; methodology, Z.Z.; software, M.X.; validation, Z.Z. and Y.Z.; formal analysis, Z.Z.; investigation, D.H.; resources, M.X.; data curation, Z.Z.; writing—original draft preparation, Q.W.; writing—review and editing, Y.Z.; visualization, Y.Z.; supervision, Y.Z.; project administration, Y.Z.; funding acquisition, Y.Z. All authors have read and agreed to the published version of the manuscript.

**Funding:** This research was funded by Inner Mongolia "science and technology" action focus on special "Research and Application of Key Technologies for Production and Processing of Durum Wheat and Products in Hetao irrigation area" (NMKJXM202111-3) and Inner Mongolia Natural Science Foundation of China "Research on nitrogen nutrition diagnosis of spring wheat in Hetao irrigation area based on UAV mapping technology" (2021MS03089).

**Data Availability Statement:** Not applicable.

**Acknowledgments:** We truly appreciate the assistance from the Wuyuan Agricultural Technology Extension Center at Bayannur, Inner Mongolia, China.

**Conflicts of Interest:** The authors declare no conflict of interest.

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
