# Peer review of "Estimation of Relative Chlorophyll Content in Spring Wheat Based on Multi-Temporal UAV Remote Sensing"

_agronomy, doi:10.3390/agronomy13010211_

Round 1

Reviewer 1 Report

Dear Authors,

the paper is presented in clear and coherent manner, the experimental design is precise and well mapped out.

1.What is the main question addressed by the research? The SPAD is a simple method to estimate chlorophyll content in leaves, the SPAD handheld meter is widely used in agriculture, so it's important to understand the correlation between other physiological index. This paper study some of the parameters.
2.Do you consider the topic original or relevant in the field? Does it address a specific gap in the field?
3.What does it add to the subject area compared with other published material? There are a lot of studies that use the SPAD meter, this study is specific on the parameters obtained with remote sensors, although the results are specific for wheat in determined phenological stages. 4.What specific improvements should the authors consider regarding the methodology? What further controls should be considered? The work is simple, clearer and linear, covers specific and determined parameters and situations, and can be published.
5.Are the conclusions consistent with the evidence and arguments presented and do they address the main question posed? The conclusions are consistent with the experiment and statistical data; however they can be extended considering the nutritional and hydric conditions.
6.Are the references appropriate? The references are enough to a complete introduction and to analysis results.
7.The authors should add the reference to fig. 7 in the text
8.Please include any additional comments on the tables and figures. As already written in my comments for authors, I suggest a better description in the caption and titles of figures and tables.

Reviewer 2 Report

Comments to Authors

The paper “Estimation of relative chlorophyll content in spring wheat based on multi-temporal UAV remote sensing” is very interesting in the context of characterize the chlorophyll content of plants in a continuous rapid, nondestructive, and accurate manner. However, there are still many errors and comments that must be taken into account before approve the manuscript for publication.  

1-      Manuscript needs native English editing

2-      Lines 10: Please change “tasseling” in this line and all the manuscript into “heading stage”. The tasseling is always used with corn crop, while the heading is used with wheat to express the expulsion of wheat spikes. 

3-      Lines 14: changes “7 d, 14 d, 21 d, and 28 d after wheat tasseling” into “ 7, 14, 21, and 28 days after heading stage”

4-      Line 17: please write the full name for DNN, PLS, RE, and Ada

5-      Line 18: what is the meaning “four different growth stages” the measurements were done at different times from heading growth stage

6-      Line 29: Most Keywords are already mentioned in the title. Please use other words, not mentioned in the title.

7-      Line 40: change “chlorophyll that leaves the living body is easily decomposed by light” into “chlorophyll that extracted from the living body is easily decomposed by light

8-      Lines 41-43: please added an additional sentence after lines 41-43 which indicates that the manual handheld chlorophyll meter provides only information on the state of the
chlorophyll content based on a single leaf and disregard the vertical heterogeneity in the chlorophyll content within a canopy.  

9-      Line 61: what the meaning “MCA”. Please write the full name.

10-  Lines 65-67: This information about wheat growth stages is not correct. There are three development stages (foundation, construction, production) of wheat. Foundation stage initiate from seed emergence to stem elongation, construction stage starts from first node detachable from flowering. It is very critical yield stage. The production phase starts from flowering to ripening. There are many scales to measure the growth stages of wheat like Feeks scale, Zadoks and Haun scales. Please check these scales.

11-  Lines 86-93: This paragraph should be moved to the M&M section

12-  Line 100: “split-zone design” There is no design with this name. The design whose name I know is split-plot design. Please check this design

13-  Line 100: change the main zone to the main plot  

14-  Line 101: Please change tiller stage to tillering stage

15-  Line 101: There is no growth stage in wheat named “plucking stage” please check the Zadoks scale for wheat and use the correct name for each stage that is mentioned in the manuscript “Zadoks, J.C.; Chang, T.T.; Konzak, C.F. A Decimal Code for the Growth Stages of Cereals. Weed Res. 1974, 14, 415–421.

16-  Line 102: Please change filling stage to grain filling stage. There are three sub-stages for grain filling stage, please identify which sub-stage was irrigated.

17-  Line 102: “nodulation” please check the different growth stages of wheat carefully. There is no growth stage in wheat named “nodulation”  

18-  Line 103: convert hm2 to ha “please use standard units for measurements in all text

19-   Line 103: change secondary zone to subplot

20-  Line 103: each time with 900 m3/hm2. This means that the amount of irrigation water for the first treatment was 3600 m3/ha (4*900) and the second treatment was 1800 m3/ha (2*900). I think this amount of water irrigation is not enough for wheat to well grow as it causes drought stress unless there is enough rain. Unfortunately, there is no information about rainfall

21-  Line 104: what is the difference between treatment CK (no fertilizer) and N0 (0 kg/hm2)

22-  Line 105: why added, “12 treatments” in this place? Please write it in a new sentence

23-  Line 106-107: sowing rate was set at 375 kg/hm2. Is this seeding rate correct for wheat? To our knowledge, the seeding rate for wheat ranged from 150 to 200 kg ha-1

24-   No information was mentioned in the M&M section about the weather data such as temperature, rainfall, and humidity.

25-  Lines 113-116, please provide a photo for Phantom 4 multispectral, P4M

26-  Lines 138-139: Table 2 should be moved to the Result section. The data of SPAD is not complete. It is important to present the SPAD values for all treatments (12 treatments) and it is also important to analyze these data statistically.

27-  Line 189: Table 3. It is very important to show the Correlation coefficient between spectral vegetation index and SPAD under each irrigation treatment.

28-  Line 306, where is Figure 6-c?

29-  Line 316: Figure 7 Violin plots of the spectral indices. There is no mention of this table in the text.

Reviewer 3 Report

See comments in the attached file
